# Steroidogenesis and androgen/estrogen signaling pathways are altered in in vitro matured testicular tissues of prepubertal mice

Laura Moutard[1], Caroline Goudin[1], Catherine Jaeger[1], Céline Duparc[1], Estelle Louiset[1], Tony Pereira[2], François Fraissinet[2], Marion Delessard[1], Justine Saulnier[1], Aurélie Rives-Feraille[1], Christelle Delalande[3], Hervé Lefebvre[1], Nathalie Rives[1], Ludovic Dumont[1], Christine Rondanino[1]*

[1]Univ Rouen Normandie, Inserm, Normandie Univ, NorDiC UMR 1239, Adrenal and Gonadal Pathophysiology team, F-76000, Rouen, France; [2]Department of General Biochemistry, Rouen University Hospital, Rouen, France; [3]Normandie Univ, UNICAEN, OeReCa, Caen, France

*For correspondence:
christine.rondanino@univ-rouen.fr

Competing interest: The authors declare that no competing interests exist.

**Abstract** Children undergoing cancer treatments are at risk for impaired fertility. Cryopreserved prepubertal testicular biopsies could theoretically be later matured in vitro to produce spermatozoa for assisted reproductive technology. A complete in vitro spermatogenesis has been obtained from mouse prepubertal testicular tissue, although with low efficiency. Steroid hormones are essential for the progression of spermatogenesis, the aim of this study was to investigate steroidogenesis and steroid signaling in organotypic cultures. Histological, RT-qPCR, western blot analyses, and steroid hormone measurements were performed on in vitro cultured mouse prepubertal testicular tissues and age-matched in vivo controls. Despite a conserved density of Leydig cells after 30 days of culture (D30), transcript levels of adult Leydig cells and steroidogenic markers were decreased. Increased amounts of progesterone and estradiol and reduced androstenedione levels were observed at D30, together with decreased transcript levels of steroid metabolizing genes and steroid target genes. hCG was insufficient to facilitate Leydig cell differentiation, restore steroidogenesis, and improve sperm yield. In conclusion, this study reports the failure of adult Leydig cell development and altered steroid production and signaling in tissue cultures. The organotypic culture system will need to be further improved before it can be translated into clinics for childhood cancer survivors.

## eLife assessment

This study reports **useful** information on the limits of the organotypic culture of neonatal mouse testes, which has been regarded as an experimental strategy that can be extended to humans in the clinical setting for the conservation and subsequent re-use of testicular tissue. The evidence that the culture of testicular fragments of 6.5-day-old mouse testes does not allow optimal differentiation of steroidogenic cells is **compelling** and should enable further optimizations in the future.

## Introduction

Pediatric cancer treatments such as chemotherapy have recognized toxicity in germline stem cells, which could lead to infertility in adulthood (*Allen et al., 2018*; *Miller et al., 2016*). Freezing of

prepubertal testicular tissue containing spermatogonia is a fertility preservation option proposed for prepubertal boys with cancer prior to highly gonadotoxic treatments. Several fertility restoration approaches, whose aim is to mature cryopreserved tissues *in vivo* or *in vitro* in order to produce spermatozoa, are being developed. So far, these approaches have not been clinically validated. *In vitro* maturation strategies are currently being optimized in animal models. *In vitro* spermatogenesis could indeed be proposed to patients with testicular localization of residual tumor cells, for whom testicular tissue autografting is not indicated (about 30% of patients with acute leukemia). The organotypic culture procedure, which preserves testicular tissue architecture, microenvironment, and cell interactions, has been used successfully to obtain spermatozoa from fresh or frozen/thawed mouse prepubertal testicular tissues (*Sato et al., 2011*; *Yokonishi et al., 2014*; *Arkoun et al., 2015*; *Dumont et al., 2015*). In addition, viable and fertile mouse offspring have been obtained from *in vitro* produced spermatozoa by oocyte microinjection (*Sato et al., 2011*; *Yokonishi et al., 2014*).

It has been previously shown that supplementing organotypic culture media with retinol improves the *in vitro* production of spermatids and spermatozoa in prepubertal mouse testicular tissues cultured at a gas-liquid interphase (*Arkoun et al., 2015*; *Dumont et al., 2016*). However, *in vitro* sperm production still remains a rare event. Transcript levels of the androgen receptor (AR)-regulated gene *Rhox5* were decreased at the end of the culture period, suggesting that testosterone production by Leydig cells and/or AR transcriptional activity was impaired in organotypic cultures (*Rondanino et al., 2017*). A decline in intratesticular testosterone levels has been highlighted *in vitro* in 35- to 49 day cultures of 5 d*pp* mouse testes, i.e., after the end of the first wave of spermatogenesis (*Pence et al., 2019*). Moreover, our recent transcriptomic analysis reveals a decline in mRNA levels of *Cyp17a1*, encoding a steroidogenic enzyme, in 4- to 30 day cultures of 6 d*pp* mouse testes (*Dumont et al., 2023*). Based on these results, Leydig cell maturation and functionality need to be thoroughly explored in order to identify the molecular mechanisms that are deregulated *in vitro*.

Two distinct populations of Leydig cells appear during mouse testicular development: (i) fetal Leydig cells arising during embryonic development and regressing over the first two weeks of postnatal life and (ii) adult Leydig cells appearing around one week after birth. Although they both express StAR (steroidogenic acute regulatory protein), CYP11A1 (P450 side chain cleavage), 3β-HSD1 (3β-hydroxysteroid dehydrogenase type 1), and CYP17A1 (P450 17α-hydroxylase/C17-20 lyase), which are steroidogenic proteins required for androgen production, 17β-HSD3 (17β-hydroxysteroid dehydrogenase type 3), which converts androstenedione to testosterone, is expressed only by adult Leydig cells (*O'Shaughnessy et al., 2002*; *Sararols et al., 2021*). Consequently, the major androgens produced in fetal Leydig cells and adult Leydig cells are androstenedione and testosterone, respectively (*O'Shaughnessy et al., 2000*). In rodents, adult Leydig cells develop from undifferentiated stem Leydig cells through two intermediate cells, progenitor Leydig cells and immature Leydig cells (*Ye et al., 2017*). All these cells, except stem Leydig cells, express CYP11A1, 3β-HSD1, and CYP17A1 (*Jiang et al., 2014*; *Zhang et al., 2004*). 17β-HSD3 begins to be expressed in immature Leydig cells (*Ge and Hardy, 1998*). The expression of the androgen metabolizing enzyme SRD5A1 (5α-reductase 1), which is high in progenitor and immature Leydig cells, is silenced in adult Leydig cells (*Ge and Hardy, 1998*). Adult Leydig cells express INSL3 (Insulin-like peptide 3) (*Balvers et al., 1998*; *Mendis-Handagama et al., 2007*) as well as SULT1E1 (estrogen sulfotransferase) (*Sararols et al., 2021*), which protects Leydig cells and seminiferous tubules against estrogen overstimulation by catalyzing the sulfoconjugation and inactivation of estrogens. Estrogens and androgens can also be inactivated in the testis by 17β-HSD2 (17β-hydroxysteroid dehydrogenase type 2), which converts estradiol to estrone and testosterone to androstenedione (*Wu et al., 1993*). Different factors, among which DHH (Desert hedgehog), IGF1 (Insulin-like Growth Factor 1), and LH (Luteinizing hormone) regulate the proliferation and differentiation of the Leydig cell lineage: DHH is required for both proliferation and differentiation of stem Leydig cells (*Li et al., 2016*), IGF1 stimulates the proliferation of progenitor Leydig cells and immature Leydig cells (*Hu et al., 2010*) and promotes the maturation of immature Leydig cells into adult Leydig cells (*Wang et al., 2003*), and LH is critical to both adult Leydig cell differentiation and their precursor proliferation (*Ma et al., 2004*).

The steroid hormones produced by Leydig cells are essential for the progression of spermatogenesis. Testosterone, synthesized under the control of LH, is essential for many aspects of spermatogenesis, including meiotic progression, spermiogenesis, and germ cell survival (*De Gendt et al., 2004*; *Chang et al., 2004*; *Holdcraft and Braun, 2004*; *O'Shaughnessy et al., 2012*). Androgen-binding

protein (ABP), produced by Sertoli cells under the regulation of FSH, increases the accumulation of androgens in the seminiferous epithelium and makes them available for binding to intracellular AR (*Hagenäs et al., 1975*). Estrogens, derived from the aromatization of androgens by CYP19A1 (aromatase) in somatic and germ cells, have been shown to be important for the long-term maintenance of spermatogenesis in the ArKO mouse (lacking aromatase) and for the progression of normal germ cell development in the ENERKI mouse (estrogen nonresponsive ERα knock-in) (*Robertson et al., 1999*; *Sinkevicius et al., 2009*). The expression of aromatase is age-dependent, being mostly in Sertoli cells in immature rat testes and then in Leydig cells during adulthood (*Rommerts et al., 1982*; *Papadopoulos et al., 1986*). In mouse, aromatase is also present within seminiferous tubules, mainly in spermatids (*Nitta et al., 1993*).

Androgens act directly on somatic cells (Sertoli cells and peritubular myoid cells) through AR to support germ cell development by regulating the expression of target genes. *Rhox5*, encoding a transcription factor that indirectly promotes germ cell survival (*MacLean et al., 2005*), and *Eppin*, encoding a serine protease inhibitor, are target genes of testosterone in Sertoli cells, with androgen response elements (ARE) in their promoters (*Willems et al., 2010*). Several estrogen receptors mediate the effects of estrogens in the testis. In rodents, ERα was found in Leydig cells and peritubular myoid cells, ERβ was found in Sertoli cells and some germ cells (spermatogonia, spermatocytes), and GPER was detected in Leydig cells, Sertoli cells, and germ cells (spermatocytes, spermatids) (*Zhou et al., 2002*; *Kotula-Balak et al., 2018*; *Lucas et al., 2010*; *Chimento et al., 2010*; *Chimento et al., 2011*). *Faah*, encoding a hydrolase that promotes Sertoli cell survival by degrading anandamide, is a direct target gene of estrogens in mature Sertoli cells (*Grimaldi et al., 2012*; *Rossi et al., 2007*). *Septin12*, containing 1 AR and 2 ERα binding sites in its promoter, has been identified as a potential novel target of the androgen and estrogen receptors in human testicular cells (*Kuo et al., 2019*).

Since steroid hormones play an essential role in the progression of spermatogenesis, it appears necessary to ensure that their syntheses and mechanisms of action are not altered in *in vitro* cultured testicular tissues. The aim of the present work was, therefore, to study Leydig cell maturation, steroidogenesis, and androgen/estrogen signaling in a comprehensive manner during the *in vitro* maturation of mouse prepubertal testicular tissues. The expression of Leydig cell markers, Leydig cell differentiation factors, steroidogenic enzymes, steroid receptors, steroid target genes, and steroid contents were quantified in cultured tissues and in their *in vivo* counterparts. Although our organotypic culture conditions support a complete spermatogenesis, a failure of adult Leydig cell development with impaired steroidogenesis and altered steroid hormone signaling was demonstrated in this study.

## Results

### Leydig cells are partially mature after 30 days of organotypic culture

We first wondered whether Leydig cell number, survival, proliferation, and differentiation could be altered under *in vitro* conditions. 3β-HSD immunofluorescence staining was performed to detect and quantify Leydig cells during mouse postnatal development and in *in vitro* cultured fresh testicular tissues (FT). The percentage of Leydig cells in apoptosis or in proliferation was measured after cleaved caspase 3 or Ki67 immunofluorescence staining respectively, and the percentage of Leydig cells expressing AR, which is required for their proliferation and maturation (*O'Shaughnessy et al., 2019*), was also determined. The transcript levels of genes necessary for Leydig cell differentiation (*Igf1*, *Dhh*) and markers of fetal Leydig cells (*Mc2r*), progenitor/immature Leydig cells (*Srd5a1*), or adult Leydig cells (*Sult1e1*, *Insl3*) were then assessed by RT-qPCR. Finally, the concentration of INSL3, a marker of Leydig cell maturity, was measured by RIA in organotypic culture media and in testicular homogenates.

After 16 days of culture, the number of Leydig cells per cm² of testicular tissue was not significantly different from 22 d*pp*, the age-matched *in vivo* control (*Figure 1A–B*). However, at this time point, the percentage of apoptotic Leydig cells was increased compared to the *in vivo* control while the percentage of proliferating Leydig cells was unchanged (*Figure 1C–D*). In addition, the percentage of Leydig cells expressing AR was similar at D16 and 22 d*pp* (*Figure 1E*). Whereas no difference was observed in the mRNA levels of *Dhh*, *Srd5a1* and *Sult1e1* between D16 and 22 d*pp*, *Igf1* and *Insl3* transcripts levels were respectively higher and lower in 16 day organotypic cultures than *in vivo*

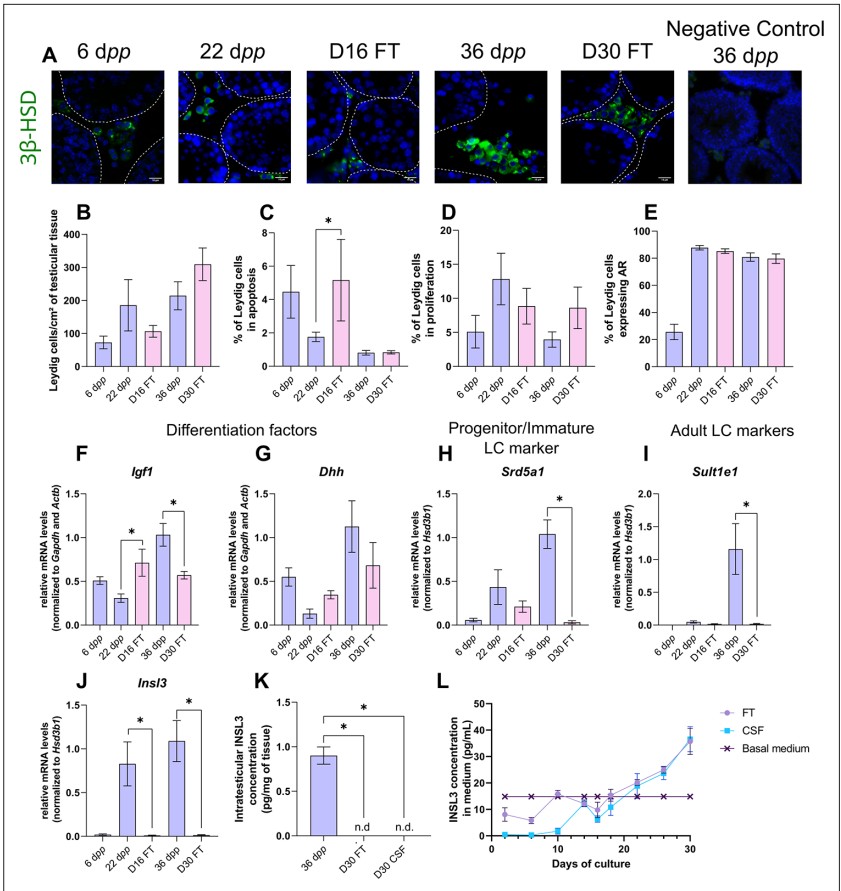

**Figure 1.** Leydig cells are partially mature after 30 days of organotypic culture. (**A**) Representative images of 3β-hydroxysteroid dehydrogenase (3β-HSD) expression by Leydig cells during mouse postnatal development (6 d*pp*, 22 d*pp,* and 36 d*pp*) and in *in vitro* cultured tissues after 16 days of culture (D16) or 30 days (D30). A representative image of a negative control, carried out by omitting the primary antibody, is also shown. Testicular tissue sections were counterstained with Hoechst (blue). Dotted lines delineate seminiferous tubules. Scale: 15 µm. (**B**) Number of 3β-HSD + Leydig cells per cm² of testicular tissue during mouse postnatal development (6 d*pp*, 22 d*pp,* and 36 d*pp*) and in *in vitro* cultured tissues (D16 and D30). (**C–D**) Percentage of Leydig cells (**C**) in apoptosis (3β-HSD and cleaved caspase 3 positive) or (**D**) in proliferation (3β-HSD and Ki67 positive) in *in vivo* and *in vitro* matured tissues. (**E**) Percentage of 3β-HSD positive Leydig cells expressing AR in *in vivo* and *in vitro* matured tissues. (**F–J**) Relative mRNA levels of Leydig cell differentiation factors (*Igf1*, *Dhh*), progenitor/immature Leydig cell (*Srd5a1*), and adult Leydig cell markers (*Sult1e1*, *Insl3*) (normalized to *Gapdh* and *Actb* or to *Hsd3b1*). (**K–L**) Intratesticular concentration of INSL3 (pg/mg of tissue, **K**) or in the culture medium (pg/mL, **L**). Data are presented as means ± SEM with n=4 biological replicates for each group. A value of *p<0.05 was considered statistically significant. n.d.: not determined (under the detection limit) FT: Fresh Tissue; CSF: Controlled Slow Freezing.

The online version of this article includes the following source data and figure supplement(s) for figure 1:

**Source data 1.** Source data of *Figure 1*.

**Figure supplement 1.** Impact of controlled slow freezing on Leydig cells in organotypic cultures.

**Figure supplement 1—source data 1.** Source data of *Figure 1—figure supplement 1*.

(*Figure 1F–J*). *Mc2r* transcripts were barely detected at D16 and 22 d*pp* as well as at later time points, thereby reflecting the low amount of fetal Leydig cells in prepubertal testicular tissues (data not shown).

After 30 days of culture, no significant difference in the number of Leydig cells per cm² of testicular tissue and in the percentages of apoptotic and proliferating Leydig cells was observed compared to 36 d*pp*, the corresponding *in vivo* time point (*Figure 1A–D*). Furthermore, the percentage of Leydig cells expressing AR was comparable after *in vitro* or *in vivo* maturation (*Figure 1E*). The transcript levels

of *Dhh* were also similar in *in vitro* and *in vivo* matured tissues at the end of the first spermatogenic wave (*Figure 1G*). However, a reduction in the mRNA levels of *Igf1*, *Srd5a1*, and the two adult Leydig cell markers (*Sult1e1*, *Insl3*) was found at D30 in comparison to 36 d*pp in vivo* controls (*Figure 1F and H–J*). Moreover, intratesticular INSL3 was below the detection limit (<10 pg/mL) in *in vitro* matured tissues (*Figure 1K*), and a significant elevation in the concentration of INSL3 was observed in culture medium from D22 to D30 for FT tissues (*Figure 1L*).

In summary, while the number and proliferation of Leydig cells are not affected by organotypic culture conditions, their differentiation into mature adult Leydig cells is impaired *in vitro*.

## Controlled slow freezing has no impact on Leydig cell density or state of differentiation before or after organotypic culture

In the clinics, testicular biopsies from prepubertal boys are frozen and stored in liquid nitrogen for later use. In order to assess the impact of freezing/thawing procedures (CSF) on Leydig cell number, survival, proliferation, and differentiation in organotypic cultures, we conducted the same analyses as above on 6 d*pp* CSF mouse testicular tissues and on *in vitro* matured 6 d*pp* CSF tissues (*Figure 1—figure supplement 1*).

CSF had no impact on the number of Leydig cells per cm² of tissue at 6 d*pp* as well as after 16 or 30 days of culture (*Figure 1—figure supplement 1A–B*). Although the percentage of apoptotic Leydig cells was lower in CSF than in FT tissues at 6 d*pp*, it was similar in CSF and FT tissues at D16 and D30 (*Figure 1—figure supplement 1C*). Furthermore, no change in the percentage of Leydig cells in proliferation or expressing AR was found after CSF (*Figure 1—figure supplement 1D–E*). Moreover, *Dhh* and *Srd5a1* mRNA levels were similar in CSF and FT tissues (*Figure 1—figure supplement 1G–H*). *Igf1* transcript levels were decreased in CSF tissues at D16 (*Figure 1—figure supplement 1F*). Furthermore, *Sult1e1* mRNA levels were higher in CSF tissues at D30 (*Figure 1—figure supplement 1I*) and *Insl3* transcript levels were higher in CSF than in FT tissues at D16 and D30 (*Figure 1—figure supplement 1J*).

Since the number and the state of differentiation of Leydig cells are rather similar in FT and CSF tissues, it can be concluded that the freezing/thawing procedures are not harmful to these cells.

## The expression of several actors of steroidogenesis is affected in organotypic cultures

As the differentiation of Leydig cells is not fully completed in organotypic cultures, we next wanted to know if actors of the steroidogenic pathway show deregulated expression *in vitro* in comparison to physiological conditions, and thus which steps of the steroid hormone biosynthesis pathway may be impaired. The transcript levels of several genes involved in steroidogenesis were measured by RT-qPCR to highlight a potential deregulation of their expression in cultured tissues (*Figure 2A–D, F H–I*). The protein levels of two steroidogenic enzymes, 3β-HSD, and CYP17A1, were also quantified by western blot (*Figure 2E and G*).

Controlled slow freezing had no impact on the mRNA levels of all the genes examined at 6 d*pp*, i.e., before culture (*Figure 2A–I*). At D16, the mRNA levels of *Star*, *Cyp17a1*, and *Hsd17b3* were decreased in FT and CSF testicular tissues compared to 22 d*pp* testes (*Figure 2B, F and H*). *Cyp11a1* transcript levels were also lower in FT tissues at D16 than at 22 d*pp* (*Figure 2C*). In contrast, *Lhcgr*, *Hsd3b1*, and *Hsd17b2* transcript levels remained unchanged at this time point (*Figure 2A, D, I*). 3β-HSD and CYP17A1 protein levels were not different between D16 and 22 d*pp* (*Figure 2E and G*).

The mRNA levels of *Cyp11a1*, *Hsd3b1*, *Cyp17a1*, and *Hsd17b2* were lower at D30 in both FT and CSF tissues than in the physiological situation (*Figure 2C–D, F, I*). Despite a significant decrease in *Hsd3b1* mRNA levels at D30, the protein levels of 3β-HSD were not significantly different after 30 days of culture and at 36 d*pp* (*Figure 2E*). Western blot experiments however showed that the expression of CYP17A1 was also reduced at the protein level in both FT and CSF tissues at D30 compared to 36 d*pp* (*Figure 2G*). This steroidogenic enzyme was still detectable by immunofluorescence in Leydig cells in cultured tissues (*Figure 2J*). Finally, *Lhcgr* mRNA levels were found significantly reduced at D30 in FT tissues (*Figure 2A*), while *Star* and *Hsd17b3* mRNA levels were reduced at D30 in CSF tissues (*Figure 2B and H*).

Thus, the expression of several genes encoding steroidogenic enzymes is decreased *in vitro*, notably that of *Cyp17a1*, necessary for the conversion of progesterone to androstenedione.

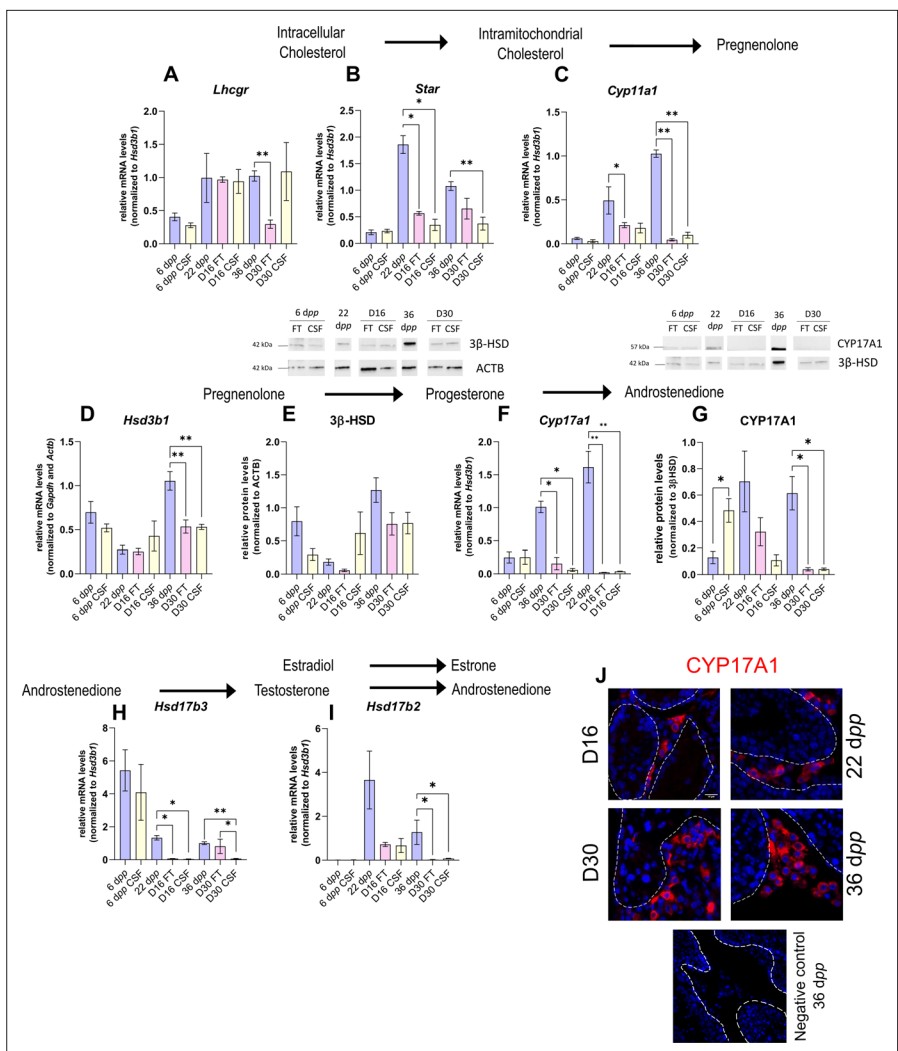

**Figure 2.** The expression of several actors of steroidogenesis is downregulated in 30 day organotypic cultures. (**A–I**) Relative mRNA levels of *Lhcgr*, *Star*, *Cyp11a1*, *Hsd3b1*, *Cyp17a1*, *Hsd17b3,* and *Hsd17b2* (normalized to *Gapdh* and *Actb* or to *Hsd3b1*) and relative protein levels of 3β-hydroxysteroid dehydrogenase (3β-HSD) (normalized to ACTB) and CYP17A1 (normalized to 3β-HSD) during mouse postnatal development (6 d*pp*, 22 d*pp*, and 36 d*pp*) and in *in vitro* cultured fresh testicular tissues (FT) or controlled slow freezing (CSF) tissues (D16 and D30). (**J**) Representative images of CYP17A1 expression during mouse postnatal development (22 d*pp* and 36 d*pp*) and in *in vitro* cultured tissues (D16 and D30). A representative image of a negative control, carried out by omitting the primary antibody, is also shown. Testicular tissue sections were counterstained with Hoechst (blue). Dotted lines delineate seminiferous tubules. Scale: 15 µm. Data are presented as means ± SEM with n=4 biological replicates for each group. A value of *p<0.05 and **p<0.01 were considered statistically significant.

The online version of this article includes the following source data for figure 2:

**Source data 1.** Source data of *Figure 2*.

## Increased intratesticular concentrations of progesterone and estradiol combined with decreased intratesticular concentration of androstenedione in organotypic cultures

We then analyzed the steroid hormone content in *in vitro* cultured testicular tissues. The concentrations of androstenedione, DHEA, and testosterone were measured by LC-MS/MS, and the concentrations of progesterone and estradiol were assessed by ELISA in both testicular samples and the culture media (*Figure 3*). DHEA was below the detection limit (<1 ng/mL) in all the samples examined (data not shown). The intratesticular concentrations of progesterone were significantly increased at D16 and

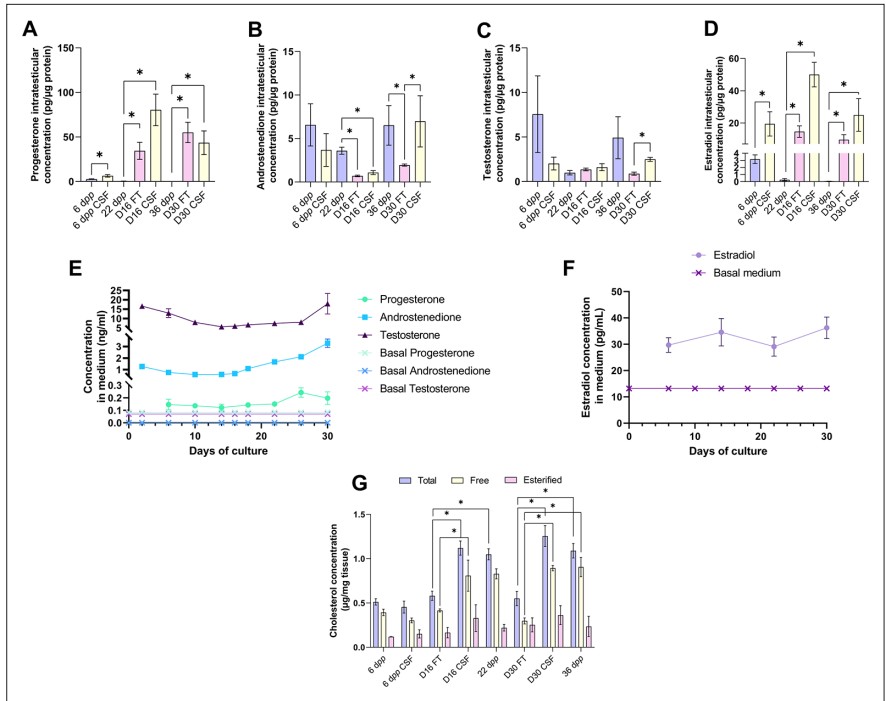

**Figure 3.** An increased production of progesterone and estradiol and a decreased production of androstenedione are observed after 16 and 30 days of culture of prepubertal mouse testicular tissues. Intratesticular concentrations of (**A**) progesterone, (**B**) androstenedione, (**C**) testosterone, and (**D**) estradiol during mouse postnatal development (6 d*pp*, 22 d*pp,* and 36 d*pp*) and in *in vitro* cultured fresh (FT) or frozen/thawed (CSF) tissues (D16 and D30). Steroid concentrations were normalized to protein levels. (**E–F**) Concentrations of (**E**) progesterone, androstenedione, testosterone, and (**F**) estradiol in the culture medium of FT tissues. (**G**) Intratesticular concentrations of total, free, and esterified cholesterol normalized to tissue mass. Data are presented as means ± SEM with n=4 biological replicates for each group. A value of *p<0.05 was considered statistically significant.

The online version of this article includes the following source data and figure supplement(s) for figure 3:

**Source data 1.** Source data of *Figure 3*.

**Figure supplement 1.** Impact of controlled slow freezing on steroids production by Leydig cells in organotypic cultures.

**Figure supplement 1—source data 1.** Source data of *Figure 3—figure supplement 1*.

D30 in both FT and CSF tissues compared to their respective *in vivo* controls (*Figure 3A*). In contrast, the intratesticular concentrations of androstenedione were significantly decreased at D16 in both FT and CSF tissues and at D30 in FT tissues compared to *in vivo* controls (*Figure 3B*). The intratesticular concentrations of testosterone were however not different in *in vitro* and *in vivo* matured tissues at both time points (*Figure 3C*). In addition, the intratesticular concentrations of estradiol were significantly higher at D16 and D30 in FT and CSF tissues than in their respective *in vivo* controls (*Figure 3D*).

In the organotypic culture medium, a significant increase in the concentrations of progesterone was observed at D26 only (*Figure 3E*). Furthermore, a significant elevation in the concentrations of androstenedione was observed from D16 to D30 (*Figure 3E*). Regarding testosterone, following a decrease in the levels of this androgen in the culture media between D2 and D14, an augmentation was then detected between D26 and D30 (*Figure 3E*). At D30, the concentrations of both androstenedione and testosterone were higher in the culture media of FT tissues than of CSF tissues (*Figure 3—figure supplement 1*). Regarding estradiol, no significant change in its concentrations was observed in the media between D6 and D30 (*Figure 3F*).

Since steroids are derived from cholesterol, intratesticular levels of total, free, and esterified cholesterol were also analyzed (*Figure 3G*). The intratesticular levels of esterified cholesterol were comparable between the different experimental conditions (*Figure 3G*). However, total cholesterol levels were higher in FT and CSF cultured tissues at D16 and D30 compared to their corresponding *in*

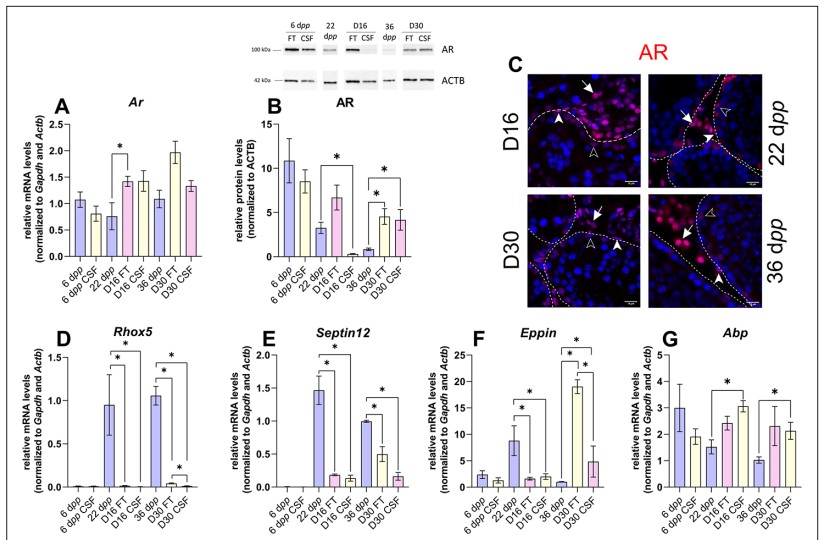

**Figure 4.** Androgen signaling is altered in 30 days organotypic cultures. (**A**) Relative mRNA levels of *Ar* (normalized to *Gapdh* and *Actb*) and (**B**) relative protein levels of androgen receptor (AR) (normalized to ACTB) during mouse postnatal development (6 d*pp*, 22 d*pp*, and 36 d*pp*) and in *in vitro* cultured fresh testicular tissues (FT) or controlled slow freezing (CSF) tissues (D16 and D30). (**C**) Representative images of AR expression at 22 d*pp* and 36 d*pp* and at corresponding *in vitro* time points (D16 and D30). A representative image of a negative control is shown in *Figure 2J* (same secondary antibody as for CYP17A1). Testicular tissue sections were counterstained with Hoechst (blue). Solid arrowheads: peritubular myoid cells. Open arrowheads: Sertoli cells. Arrows: Leydig cells. Dotted lines delineate seminiferous tubules (ST). Scale: 15 µm. (**D–G**) Relative mRNA levels of *Rhox5*, *Septin12*, *Eppin,* and *Abp* (normalized to *Gapdh* and *Actb*). Data are presented as means ± SEM with n=4 biological replicates for each group. A value of *p<0.05 was considered statistically significant.

The online version of this article includes the following source data for figure 4:

**Source data 1.** Source data of *Figure 4*.

---

*vivo* time points (*Figure 3G*). Moreover, free cholesterol levels were increased in FT and CSF cultured tissues at D30 compared to 36 d*pp* and in FT tissues at D16 (*Figure 3G*).

Overall, our data show that the steroid content is altered in cultured testicular tissues, with an excess of free cholesterol, progesterone, and estradiol and a deficiency of androstenedione.

## Androgen and estrogen signaling are altered in 30-day organotypic cultures

The impact of imbalanced steroid hormone levels on downstream target gene expression was then investigated. The mRNA levels of actors of androgen and estrogen signaling were measured by RT-qPCR (*Figures 4 and 5*). The protein expression of the AR, aromatase (CYP19A1), and fatty acid amide hydrolase (FAAH whose expression is regulated by estrogen) was also assessed by western blot (*Figures 4 and 5*).

The transcript levels of *Ar,* encoding the androgen receptor, were higher at D16 in FT tissues than at 22 d*pp* but were not significantly different between D30 and 36 d*pp* (*Figure 4A*). Conversely, AR protein levels were comparable at D16 in FT tissues and 22 d*pp* but were more elevated at D30 in both FT and CSF tissues than at 36 d*pp* (*Figure 4B*). Our immunofluorescence data showed that AR was expressed in Sertoli cells, peritubular myoid cells, and Leydig cells in 16 day and 30 day cultured tissues as well as in the corresponding *in vivo* controls (*Figure 4C*). The transcript levels of two AR-regulated genes (*Rhox5*, *Eppin*) and the AR/Erα-regulated gene *Septin12* were decreased in FT and CSF tissues cultured for 16 days compared to 22 d*pp* (*Figure 4D–F*). At D30, *Rhox5* and *Septin12* mRNA levels were still lower *in vivo* while *Eppin* mRNA levels were higher than at 36 d*pp* (*Figure 4D–F*). In addition, a significant increase in *Abp* mRNA levels was found at D16 and D30 in CSF but not in FT tissues (*Figure 4G*).

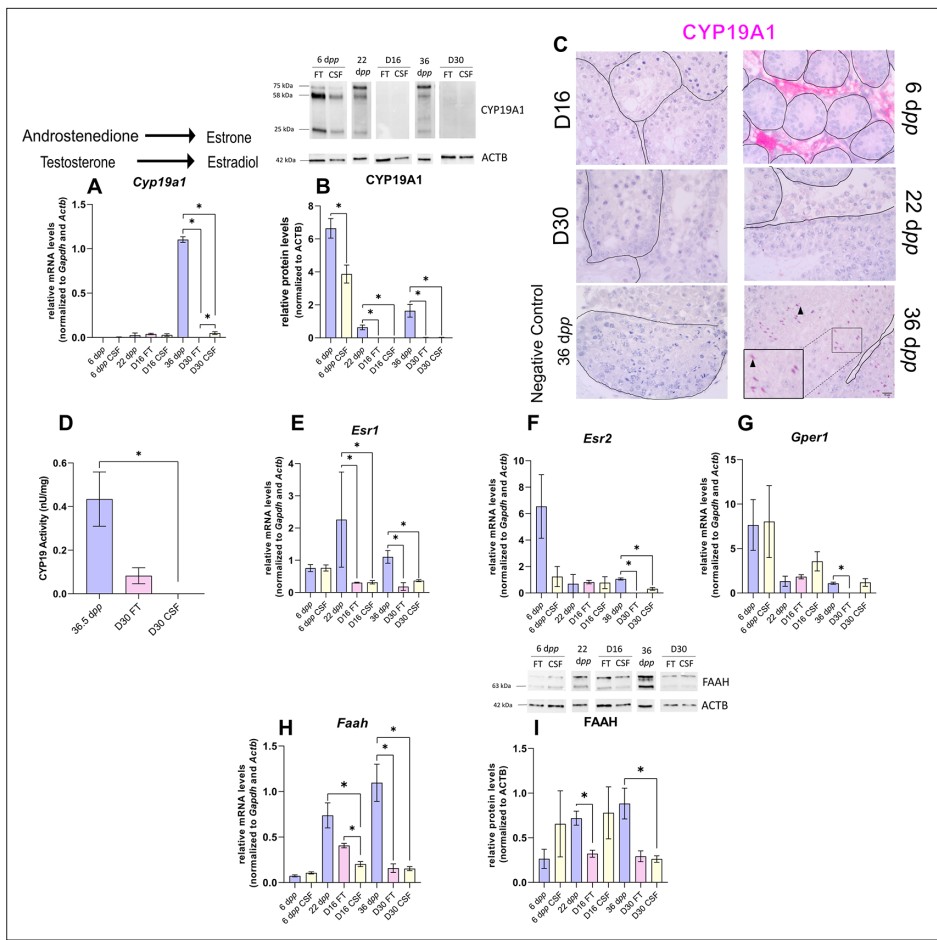

**Figure 5.** The expression of aromatase and estrogen signaling is impaired after 30 days of organotypic culture. (**A**) Relative mRNA levels of *Cyp19a1* (normalized to *Gapdh* and *Actb*) and (**B**) relative protein levels of CYP19A1 (normalized to ACTB) during mouse postnatal development (6 d*pp*, 22 d*pp*, and 36 d*pp*) and in *in vitro* cultured fresh testicular tissues (FT) or controlled slow freezing (CSF) tissues (D16 and D30). The bands on the western blot correspond to different isoforms of CYP19. (**C**) Representative images of CYP19A1 expression at 6 d*pp*, 22 d*pp*, and 36 d*pp* and at corresponding *in vitro* time points (D16 and D30). A representative image of a negative control, carried out by omitting the primary antibody, is also shown. Testicular tissue sections were counterstained with Hematoxylin. Scale: 15 μm. Arrow: Leydig cells. Arrowheads: elongated spermatids. (**D**) Aromatase activity (normalized to tissue weight). (**E–G**) Relative mRNA levels of *Esr1*, *Esr2,* and *Gper1* (normalized to *Gapdh* and *Actb*). (**H**) Relative mRNA levels of *Faah* (normalized to *Gapdh* and *Actb*) and (**I**) relative protein levels of fatty acid amide hydrolase (FAAH) (normalized to ACTB). The second band at 80 kDa is an isoform of FAAH (Q8BRM1, UniProtKB). Data are presented as means ± SEM with n=4 biological replicates for each group. A value of *p<0.05 was considered statistically significant.

The online version of this article includes the following source data for figure 5:

**Source data 1.** Source data of *Figure 5*.

Intriguingly, the expression of *Cyp19a1*, encoding aromatase, was drastically diminished after 30 days of culture in FT and CSF tissues compared to 36 d*pp* (*Figure 5A*). The protein levels of CYP19A1 were reduced in both FT and CSF tissues at D16 and D30 (*Figure 5B*). CYP19A1 was expressed in Leydig cells at 6 d*pp* and in elongated spermatids and spermatozoa at 36 d*pp*, whereas no expression could be detected in cultured testes (*Figure 5C*). CYP19A1 enzymatic activity was also significantly decreased in CSF cultures at D30 (*Figure 5D*). In addition, the transcript levels of the three genes encoding estrogen receptors (*Esr1*, *Esr2*, *Gper1*) and of the estrogen target gene *Faah* were significantly lower at D30 than at 36 d*pp* (*Figure 5E–H*). *Esr1* mRNA levels were also decreased at D16 (*Figure 5E*). FAAH protein levels were decreased in 16 day FT and 30 day CSF organotypic cultures (*Figure 5I*).

Thus, the expression of many genes in the androgen and estrogen signaling pathways are deregulated under organotypic culture conditions.

## hCG is insufficient to stimulate Leydig cell differentiation and to restore steroidogenesis and steroid hormone signaling

As LH is known to stimulate the differentiation of the adult Leydig cell lineage and steroidogenesis, we wondered whether supplementation of organotypic culture media with hCG (LH analog) could restore steroidogenesis to control levels. In order to determine the dose to be used, we examined the impact of different hCG concentrations (5 pM, 50 pM, 1 nM, 5 nM, or 50 nM) on the intratesticular concentrations of testosterone and the secretion of this hormone in the organotypic culture medium. Since the concentration of hCG leading to the best response (i.e. plateau of the dose-response curves for intratesticular and secreted testosterone) was 1 nM, we used this dose for the rest of the analyses (*Figure 6—figure supplement 1*).

Upon supplementation with 1 nM hCG, intratesticular androstenedione levels were significantly increased at D30 in FT but not CSF tissues (*Figure 6A*). The intratesticular testosterone levels were higher at D30 in both FT and CSF tissues following hCG supplementation (*Figure 6B*). Moreover, a significant increase in androstenedione and testosterone concentrations was observed in the culture media of both FT and CSF tissues following hCG supplementation (*Figure 6C–D*). The amounts of androgens released into the culture media were, however, lower for CSF tissues than for FT tissues (*Figure 6C–D*).

The addition of 1 nM hCG had no impact on the transcript levels of *Star*, *Hsd17b3*, *Hsd3b1*, *Ar*, and *Esr1* in both FT and CSF cultures (*Figure 6F, I, J, L and P*). *Hsd17b3* and *Esr1* mRNA levels still remained lower upon hCG supplementation than *in vivo* (*Figure 6I and P*). No effect of hCG was also observed in FT cultures for *Cyp17a1*, *Faah*, *Srd5a1*, *Sult1e1*, *Insl3*, *Igf1*, *Dhh*, *Abp*, and *Hsd17b2* mRNA levels (*Figure 6I, J and S*, *Figure 6—figure supplement 2*). *Lhcgr*, *Cyp11a1*, *Cyp19a1*, *Esr2*, and *Gper1* mRNA levels were increased after hCG supplementation in FT cultures (*Figure 6E, G–H, K, Q and R*). Whereas *Cyp11a1* and *Cyp19a1* mRNA levels were still lower after hCG supplementation than *in vivo* (*Figure 6G and K*), *Esr2* and *Gper1* transcripts were restored to their physiological levels (*Figure 6Q–R*). In contrast, mRNA levels of *Rhox5*, *Eppin*, *Septin12,* and *Igf1* were decreased after hCG supplementation in FT cultures, with *Rhox5* and *Septin12* transcript levels lower than those observed *in vivo* (*Figure 6M–O*).

Finally, the mean proportions of seminiferous tubules containing round and elongated spermatids were lower in tissues cultured with than without hCG (*Figure 6T*). The number of Sertoli cells inside seminiferous tubules was unchanged following hCG supplementation, but was higher than in *in vivo* controls (*Figure 6U*).

Although the addition of hCG increased androgen levels, the expression of adult Leydig cell markers, of several steroidogenic genes, and of steroid hormone-regulated genes remained low, thus not promoting the progression of *in vitro* spermatogenesis.

## Discussion

The present study shows for the first time and in a comprehensive manner that Leydig cell maturation and functionality as well as steroid hormone signaling are impaired in cultures of fresh and frozen/thawed immature mouse testes. Only a few differences were found between fresh and frozen/thawed *in vitro* matured tissues.

A similar density of Leydig cells was found in organotypic cultures and in corresponding *in vivo* controls. However, Leydig cells only partially matured *in vitro*. At D16, the expression of the Leydig cell differentiation factors *Igf1* and *Dhh* was unaffected. At D30, the proliferation/apoptosis balance was undisturbed in Leydig cells while the expression of *Insl3* was faint, as well as that of *Sult1e1*, another Leydig cell maturity marker. The deficient Leydig cell maturation observed at D30 could be related to the decreased levels of *Igf1*, which encodes insulin-like growth factor 1 known to promote the maturation of immature Leydig cells into adult Leydig cells (*Wang et al., 2003*).

Leydig cell functionality was also altered in organotypic cultures. Indeed, the expression of several genes encoding steroidogenic enzymes (*Cyp11a1*, *Cyp17a1*, *Hsd17b3*, *Hsd17b2*) was downregulated in these cells (*Figure 7*). Reduced *Cyp17a1* mRNA levels in organotypic cultures were also recently

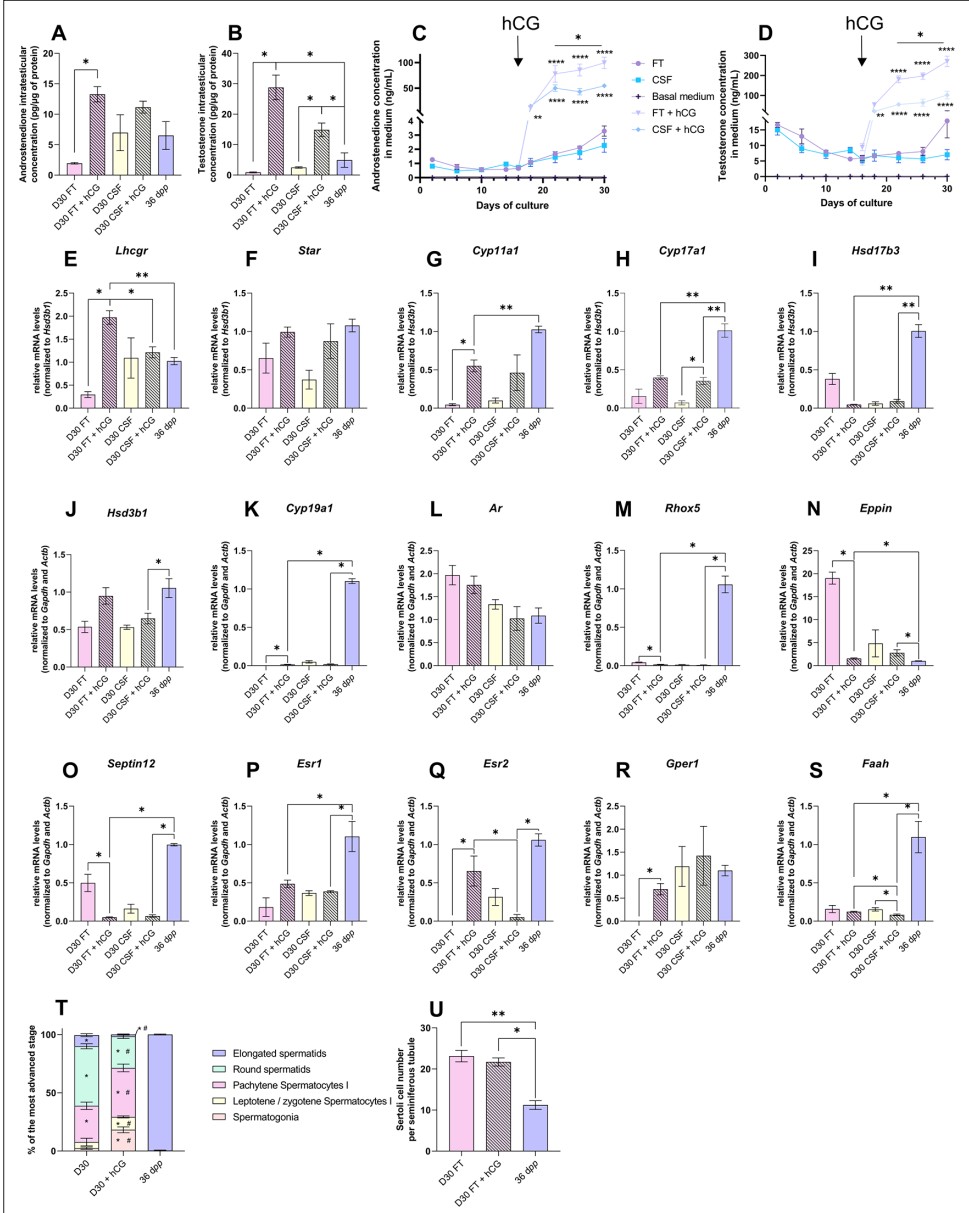

**Figure 6.** Human chorionic gonadotropin (hCG) is insufficient to stimulate Leydig cell differentiation and restore steroidogenesis and steroid hormone signaling. (**A–D**) Levels of androgens (androstenedione and testosterone) (**A–B**) in testicular tissues and (**C–D**) in culture medium with 1 nM hCG or without supplementation. (**E–S**) Relative mRNA levels of genes involved in steroidogenesis and androgen/estrogen signaling pathways (normalized to *Gapdh* and *Actb* or to *Hsd3b1*) with or without hCG supplementation. (**T**) Proportion of seminiferous tubules containing the most advanced type of germ cells after *in vitro* culture with or without hCG. p<0.05: * *vs* 36 d*pp* and # *vs* D30 FT. (**U**) Sertoli cell number in seminiferous tubules after *in vitro* culture with or without hCG. Data are presented as means ± SEM with n=4 biological replicates for each group. A value of *p<0.05, **p<0.01 and ****p<0.0001 were considered statistically significant. FT: Fresh Tissue; CSF: Controlled Slow Freezing.

The online version of this article includes the following source data and figure supplement(s) for figure 6:

**Source data 1.** Source data of *Figure 6*.

**Figure supplement 1.** Impact of different human chorionic gonadotropin (hCG) concentrations on testosterone production in organotypic cultures.

**Figure supplement 1—source data 1.** Source data of *Figure 6—figure supplement 1*.

**Figure supplement 2.** Impact of human chorionic gonadotropin (hCG) supplementation on mRNA levels of genes involved in steroidogenesis and androgen signaling in organotypic cultures.

*Figure 6 continued on next page*

*Figure 6 continued*

**Figure supplement 2—source data 1.** Source data of *figure 6 – figure supplement 2*.

highlighted using a bulk RNA-seq approach (***Dumont et al., 2023***). Decreased *Cyp11a1* and *Cyp17a1* transcript levels were previously reported in *Igf1⁻/⁻* mice, together with decreased intratesticular testosterone levels (***Hu et al., 2010***). Furthermore, deficiency in insulin-like growth factors signaling in *Insr⁻/⁻/Igf1r⁻/⁻* mouse Leydig cells was shown to impair testicular steroidogenesis (decreased *Lhcgr*, *Star*, *Cyp11a1*, *Cyp17a1*, *Hsd17b3*, *Srd5a1*, and *Insl3* mRNA levels) and increase estradiol production (***Radovic Pletikosic et al., 2021***). Here, we report for the first time an accumulation of estradiol in *in vitro* cultured tissues, which could be deleterious for sperm production. To promote Leydig cell maturation and lower intratesticular estradiol levels *in vitro*, supplementation of culture medium with IGF1 could thus be later envisaged, and even more so since this factor has been shown to increase, although modestly, the percentages of round and elongated spermatids in cultured mouse testicular fragments (***Yao et al., 2017***). To reduce estrogen levels and increase sperm production, supplementation of the organotypic culture medium with selective estrogen receptor modulators such as tamoxifen or with aromatase inhibitors such as letrozole could also be considered in future studies. These molecules have indeed proven to be effective in increasing sperm concentration in infertile men (***Cannarella et al., 2019***; ***Kooshesh et al., 2020***; ***Shuling et al., 2019***).

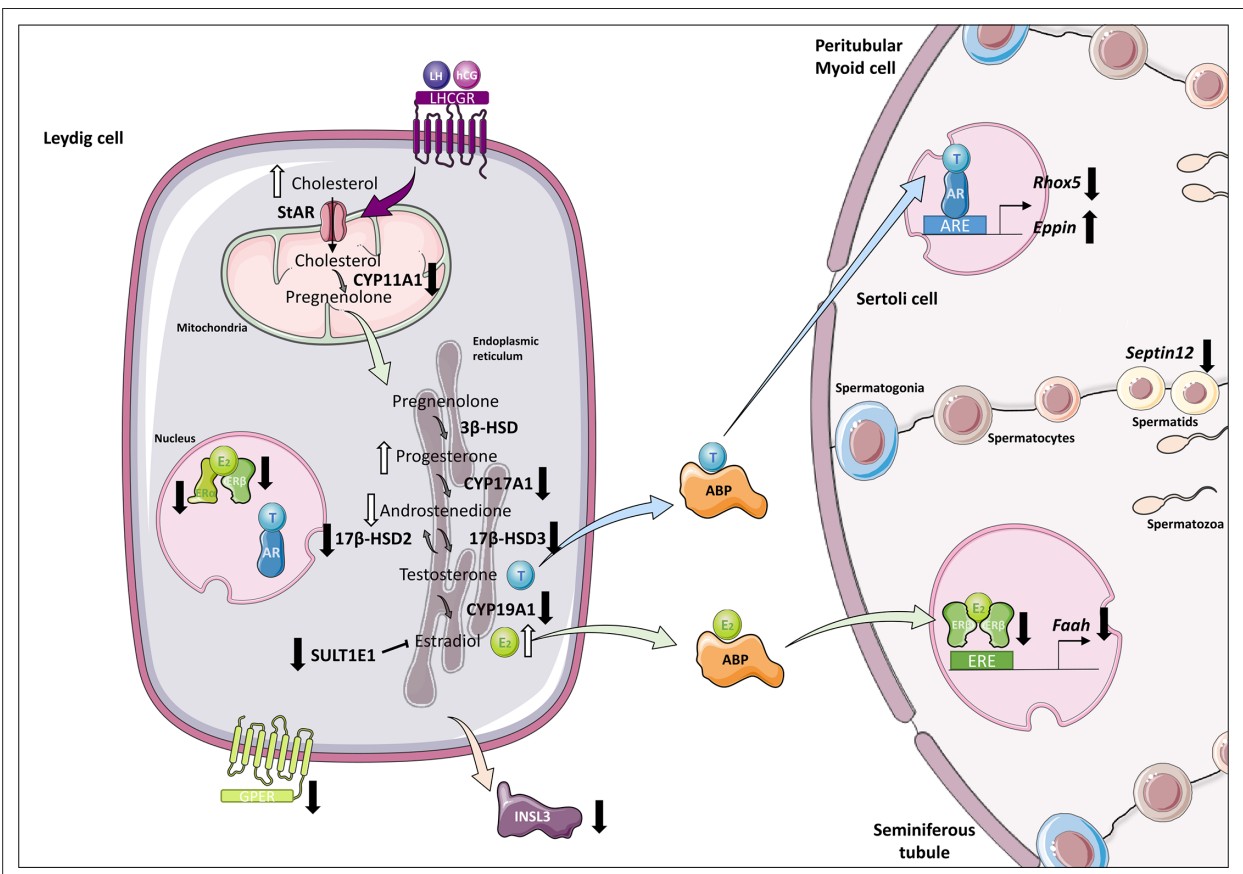

**Figure 7.** Altered steroidogenesis and androgen/estrogen signaling pathways in *in vitro* matured testicular tissues of prepubertal mice. A similar density of Leydig cells is found after 30 days of organotypic culture (D30) and at 36 days *postpartum*, the corresponding *in vivo* time point. However, Leydig cells are partially mature *in vitro* with a decrease in *Sult1e1* and *Insl3* mRNA levels (adult Leydig cell markers). The mRNA levels of *Cyp11a1*, *Cyp17a1*, and *Hsd17b3* encoding steroidogenic enzymes and the protein levels of CYP17A1 are decreased *in vitro*. Increased amounts of cholesterol, progesterone and estradiol, and decreased androstenedione intratesticular levels are observed at D30. Furthermore, despite testosterone levels similar to *in vivo*, the expression of the androgen receptor (AR) and of the androgen binding protein (*Abp*), androgen signaling is altered at D30, with decreased transcript levels of the androgen target gene *Rhox5* and of *Septin12*. Moreover, with decreased expression and activity of aromatase and decreased estrogen receptor expression, estrogen signaling is impaired at D30, leading to decreased transcript and protein levels of the estrogen target gene *Faah*.

Our work also revealed an elevation in progesterone and a reduction in androstenedione in *in vitro* matured tissues (*Figure 7*), which could arise from the reduced expression of *Cyp17a1*. A greater sperm production has been highlighted in PRKO mice (lacking the progesterone receptor), thereby showing the inhibitory action of progesterone on spermatogenesis (*Lue et al., 2013*). The cumulative excess of progesterone and estradiol in cultured testicular tissues may, therefore, slow the progression of *in vitro* spermatogenesis and lead to a poor sperm yield.

Despite the disturbed steroidogenic activity of Leydig cells *in vitro*, the accumulation of estradiol and progesterone, the decrease in androstenedione, and in contradiction with a previous study (*Pence et al., 2019*), intratesticular testosterone levels in cultured tissues were not significantly different from *in vivo*. Moreover, a complete *in vitro* spermatogenesis was achieved, albeit with a low efficiency as previously described (*Arkoun et al., 2015*; *Dumont et al., 2015*). We also found that the percentage of Leydig cells expressing AR was unchanged under *in vitro* conditions. The increased AR protein levels at D30 could thus be the result of the higher number of Sertoli cells within seminiferous tubules. Intriguingly, common points can be observed in our organotypic cultures and in LCARKO (Leydig cell-specific *Ar*^-/-) mice (*O'Hara et al., 2015*): decreased *Insl3*, *Cyp17a1,* and *Hsd17b3* mRNA levels, increased intratesticular progesterone levels and unchanged intratesticular testosterone levels compared to controls. This could suggest that AR signaling is dysfunctional in Leydig cells *in vitro*. The downregulation of *Rhox5* expression (*Figure 7*), which confirms our previous findings (*Rondanino et al., 2017*), further suggests that AR signaling may also be impaired in Sertoli cells. Recently, in a mouse model with disrupted dimerization of the AR ligand-binding-domain, a decreased expression of *Hsd17b3* and of AR-regulated genes, such as *Rhox5* and *Insl3*, a higher percentage of Sertoli cells and a decreased production of round and elongated spermatids were found (*El Kharraz et al., 2021*), which further supports our hypothesis that AR signaling may be defective in organotypic cultures. In the SPARKI (SPecificity-affecting AR KnockIn)-AR mouse model, which has a defect in binding to AR-specific DNA motifs while displaying normal interaction with the classical AREs, it has been shown that *Rhox5* expression is regulated by an AR-selective ARE, while *Eppin* expression is regulated by a classical, nonselective ARE (*Schauwaers et al., 2007*). The differential regulation of these two androgen-responsive genes could explain why *Rhox5* mRNA levels were downregulated while *Eppin* mRNA levels were upregulated in our cultures. We also report here that the transcript levels of *Septin12*, a potential target of the androgen and estrogen receptors, were reduced *in vitro* (*Figure 7*).

Additionally, estrogen signaling was impaired in organotypic cultures, with a low expression of the estrogen-synthesizing enzyme aromatase, of estrogen receptors and of the estrogen target gene *Faah* (*Figure 7*). Since the expression of *Cyp19a1*, *Esr1,* and *Esr2* (encoding aromatase, ERα, and ERβ, respectively) can be downregulated by estradiol in the rat testis (*Zanatta et al., 2021*; *Genissel et al., 2001*), we hypothesize that the transcription of these genes may be negatively controlled by elevated estradiol levels in our tissue cultures. *Faah*, whose promoter activity engages ERβ and the histone demethylase LSD1, is a direct target gene of estrogens in Sertoli cells (*Grimaldi et al., 2012*). The *Faah* proximal promoter is not regulated by estrogen in immature Sertoli cells, as they express ERβ at the same level as mature cells but do not express LSD1 (*Grimaldi et al., 2012*). Thus, the decreased *Faah* transcript levels in organotypic cultures could be the consequence of the downregulated ERβ expression and/or of the immaturity of Sertoli cells. Whether Sertoli cells in organotypic cultures are as mature as *in vivo* is unknown and will be the focus of future studies. The transcription of *Insl3*, which is repressed by estradiol in an MA-10 Leydig cell line model (*Laguë and Tremblay, 2009*), could also be inhibited by the excess of estradiol in our tissue cultures. Interestingly, our *in vitro* cultured testes exhibit common features with the testes of *Sult1e1*^-/- mice lacking estrogen sulfotransferase, an enzyme that catalyzes the sulfoconjugation and inactivation of estrogens (*Song, 2007*): decreased *Cyp17a1* expression as well as increased progesterone levels and local estrogen activity (*Qian et al., 2001*; *Tong et al., 2004*). In organotypic cultures, estrogen excess could be explained by low *Sult1e1* expression. In addition, reduced transcript levels of *Hsd17b2*, encoding an enzyme that converts estradiol to estrone and testosterone to androstenedione, may also explain why estradiol levels remain elevated in cultures while testosterone levels are similar to controls and androstenedione levels are low.

Finally, as previously reported (*Arkoun et al., 2015*; *Dumont et al., 2015*), the presence of LH or its analog hCG was not necessary to reconstitute a full first spermatogenic wave *in vitro*. However,

plasma LH concentration is known to rise *in vivo* from 21 to 28 d*pp* in the mouse model (**Wu et al., 2010**). We previously showed that the addition of 0.075 nM (50 IU/L) hCG together with FSH from D7 onwards modestly increased the number of spermatozoa produced in organotypic cultures (**Arkoun et al., 2015**). To mimic as closely as possible the physiological conditions, 1 nM hCG was added in the culture medium from D16 onwards. Even though Leydig cells responded to this stimulation by synthesizing and secreting more androgens, supplementation of the culture medium with 1 nM hCG alone was not sufficient to facilitate Leydig cell differentiation, restore steroidogenesis and steroid hormone signaling and increase the yield of *in vitro* spermatogenesis.

In conclusion, the present study shows the partial maturation and the disturbed steroidogenic activity of Leydig cells, the abnormal steroid hormone content as well as the altered androgen and estrogen signaling in organotypic cultures of fresh and frozen/thawed prepubertal mouse testicular tissues. Altogether, these defects could contribute to the low efficiency of *in vitro* spermatogenesis. It will, therefore, be necessary to optimize the culture medium by adding factors that promote proper Leydig cell differentiation as well as the culture design to prevent central necrosis from affecting the survival and/or the growth and/or the differentiation of the testis in culture. The development of an optimal model of *in vitro* spermatogenesis could be useful in advancing the field of fertility restoration, but could also have wider applications, as it could be useful for studying the initiation of puberty as well as the impact of cancer therapies, drugs, chemicals, and environmental agents (e.g. endocrine disruptors) on the developing testis.

# Materials and methods

**Key resources table**

| Reagent type (species) or resource | Designation | Source or reference | Identifiers | Additional information |
|---|---|---|---|---|
| Biological sample (*Mus musculus*, male) | Testis | CD-1 Mice from Charles River | | Freshly isolated from male *Mus musculus* (CD-1) |
| Antibody | Anti-3β-HSD (mouse monoclonal) | Santa Cruz Biot. | sc-515120 (AF488) | IF (1:100), WB (1:1000) |
| Antibody | Anti-β-Actin (mouse monoclonal) | Abcam | ab8226 | WB (1:5000) |
| Antibody | Anti-AR (rabbit monoclonal) | Abcam | ab133273 | IF (1:100), WB (1:5000) |
| Antibody | Anti-CC3 (rabbit polyclonal) | Abcam | ab49822 | IF (1:200) |
| Antibody | Anti-CYP17A1 (rabbit polyclonal) | Abcam | ab231794 | IF (1:200), WB (1:10000) |
| Antibody | Anti-CYP19A1 (mouse monoclonal) | BioRad | MCA2077S | IHC (1:50), WB (1:250) |
| Antibody | Anti-FAAH (rabbit polyclonal) | Proteintech | 17909–1-AP | WB (1:1000) |
| Antibody | Anti-Ki67 (rabbit monoclonal) | Abcam | ab16667 | IF (1:100) |
| Antibody | Anti-mouse Alexa 488 (goat) | Abcam | ab150113 | IF (1:200) |
| Antibody | Anti-rabbit Alexa 488 (goat) | Abcam | ab150077 | IF (1:200) |
| Antibody | Anti-rabbit Alexa 594 (goat) | Abcam | ab150080 | IF (1:200) |
| Antibody | Anti-mouse HRP (goat) | Invitrogen | 31430 | WB (1:5000) |
| Antibody | Anti-rabbit HRP (goat) | Invitrogen | A16110 | WB (1:5000) |
| Commercial assay or kit | RNeasy Micro kit | Qiagen | 74004 | |
| Commercial assay or kit | qScript cDNA SuperMix | QuantaBio | 95048 | |
| Commercial assay or kit | Lipid Extraction kit | Abcam | ab211044 | |
| Commercial assay or kit | Cholesterol/Cholesteryl Ester assay kit | Abcam | ab65359 | |
| Commercial assay or kit | Progesterone ELISA kit | Cayman Chemical Company | 582601 | |
| Commercial assay or kit | Estradiol ELISA kit | Cayman Chemical Company | 501890 | |
| Commercial assay or kit | Aromatase (CYP19A) Activity assay kit | Abcam | ab273306 | |

*Continued on next page*

*Continued*

| Reagent type (species) or resource | Designation | Source or reference | Identifiers | Additional information |
| --- | --- | --- | --- | --- |
| Commercial assay or kit | INSL3 RIA kit | Phoenix Pharmaceuticals | RK-035–27 | |
| Chemical compound, drug | KSR | Gibco by Life Technologies | 10828010 | (10%) |
| Chemical compound, drug | Retinol | Sigma-Aldrich | R7632 | 1 µM |
| Chemical compound, drug | hCG | MSD France | Ovitrelle | 1 nM |
| Chemical compound, drug | SYBR Green | Thermo Fisher Scientific | 4385616 | |

## Mice and testis collection

CD-1 mice (Charles River Laboratories, L'Arbresle, France) were housed in a temperature-controlled room (22–23°C) under a 12 hr light/dark cycle. Prepubertal 6 day *postpartum* (d*pp*) male mice were euthanized by decapitation and underwent a bilateral orchiectomy. Testes were transferred to Petri dishes containing α-MEM without phenol red (Gibco by Life Technologies, Saint-Aubin, France) and the complete removal of the tunica albuginea was performed with two needles under a binocular magnifier (S8AP0, Leica Microsystems GmbH, Wetzlar, Germany). Testes were then either cultured immediately (culture from fresh tissues), or after a freezing/thawing cycle (*Figure 8*). Moreover, mice aged 22 and 36 d*pp* were euthanized by $CO_2$ asphyxiation and their testes were used as *in vivo* controls for 16 and 30 days of culture, respectively (*Figure 8*).

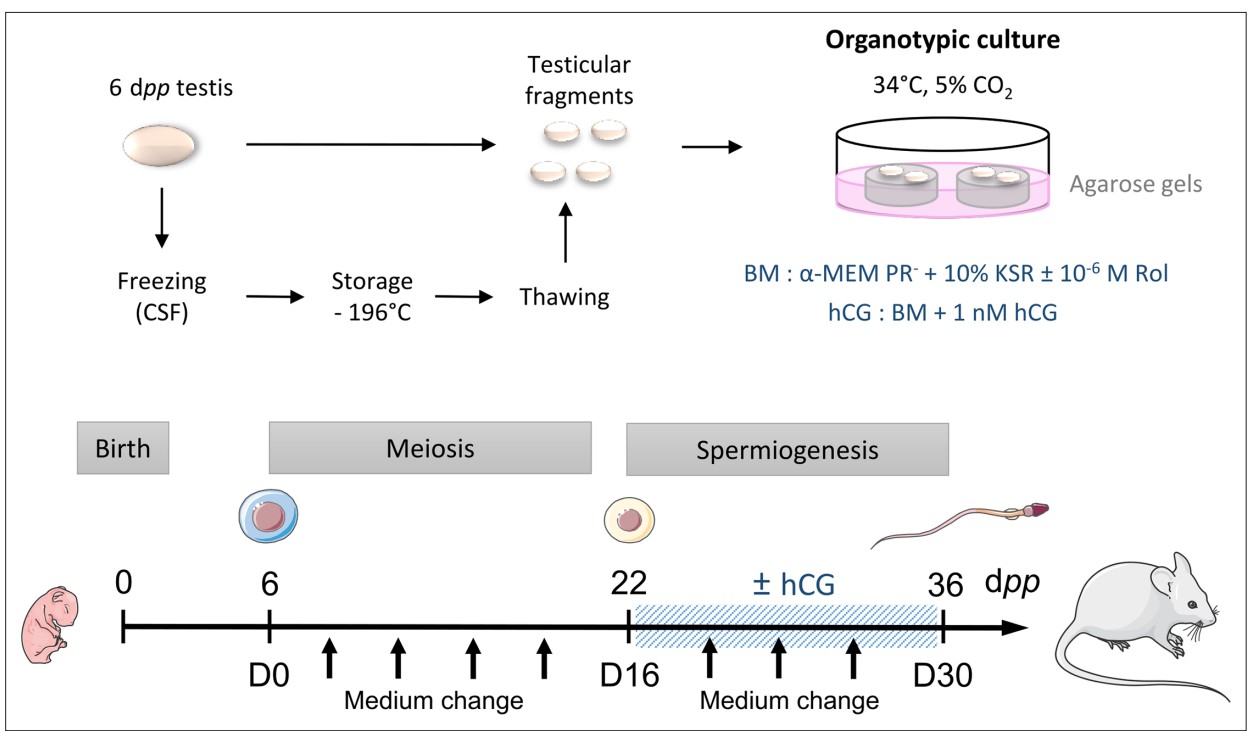

**Figure 8.** Scheme of the study design. At 6 d*pp* (D0), the testis contains only spermatogonia and the initiation of meiosis occurs between 7 and 9 d*pp*. At 22 d*pp* (D16), meiosis ends and the first round of spermatids appear. At 36 d*pp* (D30), the first spermatozoa appear and this is the end of the first spermatogenic wave. Arrows represent times at which the medium was collected (D0, D2, D6, D10, D14, D18, D22, D26, and D30). BM: Basal Medium; CSF: Controlled Slow Freezing; d*pp*: days *postpartum*; hCG: human Chorionic Gonadotropin; KSR: KnockOut Serum replacement; PR-: without phenol red; Rol: retinol.

The online version of this article includes the following source data and figure supplement(s) for figure 8:

**Figure supplement 1.** Morphological analysis of tissue sections after 16 or 30 days of organotypic cultures.

**Figure supplement 1—source data 1.** Source data of *Figure 8—figure supplement 1*.

## Controlled slow freezing (CSF) and thawing of testicular tissues

### Freezing procedure

Testes were placed into cryovials (Dominique Deutscher, Brumath, France) containing 1.5 mL of the following cryoprotective medium: Leibovitz L15 medium (Eurobio, Courtabœuf, France) supplemented with 1.5 M dimethylsulfoxide (DMSO, Sigma-Aldrich, Saint-Quentin Fallavier, France), 0.05 M sucrose (Sigma-Aldrich), 10% (v/v) fetal calf serum (FCS, Life Technologies) and 3.4 mM vitamin E (Sigma-Aldrich) (*Milazzo et al., 2008*). After a 30 min equilibration at 4 °C, samples were frozen in a programmable freezer (Nano Digitcool, CryoBioSystem, L'Aigle, France) with a CSF protocol: start at 5 °C, then –2 °C/min until reaching –9 °C, stabilization at –9 °C for 7 min, then –0.3 °C/min until –40 °C and –10 °C/min down to –140 °C. Testicular tissues were then plunged and stored in liquid nitrogen.

### Thawing procedure

Cryotubes were warmed for 1 min at room temperature (RT) and then for 3 min in a water bath at 30 °C. They were then successively incubated at 4 °C in solutions containing decreasing concentrations of cryoprotectants for 5 min each [solution 1: 1 M DMSO, 0.05 M sucrose, 10% FCS, 3.4 mM vitamin E, Leibovitz L15; solution 2: 0.5 M DMSO, 0.05 M sucrose, 10% FCS, 3.4 mM vitamin E, Leibovitz L15; solution 3: 0.05 M sucrose, 10% FCS, 3.4 mM vitamin E, Leibovitz L15; solution 4: 10% FCS, 3.4 mM vitamin E, Leibovitz L15].

### Organotypic cultures at a gas-liquid interphase

*In vitro* tissue cultures were performed as previously described (*Sato et al., 2011*; *Arkoun et al., 2015*). Briefly, prepubertal 6 day old mouse testes, which contain spermatogonia as the most advanced type of germ cells, were first cut into four fragments (approximately 0.75 mm$^3$ each, which was previously determined to be the most appropriate size for mouse *in vitro* spermatogenesis) (*Dumont et al., 2016*). They were placed on top of two 1.5% (w/v) agarose Type I gels (Sigma-Aldrich) half-soaked in a medium. Testicular tissues were then cultured under 5% $CO_2$ at 34 °C for 16 days (D16), which corresponds to the end of meiosis and the appearance of the first round of spermatids, or 30 days (D30) to explore the end of the first spermatogenic wave. No growth of the fragments was observed during the organotypic culture period (*Figure 8—figure supplement 1A*). The basal medium (BM) contained α-MEM without phenol red, 10% KSR (KnockOut Serum Replacement, Gibco by Life Technologies), 0.1 mg/mL streptomycin, and 100 UI/mL penicillin (Sigma-Aldrich). The medium was chosen without phenol red because of its estrogen activity (*Berthois et al., 1986*). Retinol ($10^{-6}$ M, Sigma-Aldrich) was added in all organotypic cultures from D2 and then every 8 days in order to respect the meiosis entry cycle of spermatogonia (*Arkoun et al., 2015*). Furthermore, the basal medium was supplemented or not with 1 nM hCG (human Chorionic Gonadotropin, MSD France, Courbevoie, France) from D16 in order to stimulate Leydig cell differentiation and assess Leydig cell functionality. Media were prepared just before use and were replaced twice a week.

## Histological analyses

### Tissue fixation, processing, and sectioning

Testicular tissues were fixed with Bouin's solution (Sigma-Aldrich) or 4% paraformaldehyde (PFA, Sigma-Aldrich) for 2 hr at room temperature. They were then dehydrated in ethanol in the Citadel 2000 tissue processor (Shandon, Cheshire, UK) and embedded in paraffin. Tissue sections (3 μm thick) were prepared with the RM2125 RTS microtome (Leica) and were mounted on Polysine slides (Thermo Fisher Scientific, Waltham, MA, USA).

### Periodic acid schiff (PAS) reaction

A PAS reaction was then performed on *in vitro* matured tissues. Tissue sections were deparaffinized in xylene and rehydrated in decreasing concentrations of ethanol. Slides were then immersed in 0.5% periodic acid (Thermo Fisher Scientific) for 10 min, rinsed for 5 min in water, and then placed for 30 min in the Schiff reagent (Merck, Darmstadt, Germany). After three 2 min washes in water, sections were counterstained with Mayer's hematoxylin and mounted with Eukitt (CML, Nemours, France). Images were acquired on a DM4000B microscope (Leica) at a 400x magnification. The

most advanced type of germ cells present in the testicular fragments was analyzed in at least 30 cross-sectioned seminiferous tubules (located at the periphery of the cultured fragments, i.e., outside of the necrotic region) (*Figure 8—figure supplement 1E–F*) from two sections with the Application Suite Core v2.4 software (Leica).

## Immunofluorescence staining

After deparaffinization, rehydration and a 3 min wash in phosphate-buffered saline (PBS, Sigma-Aldrich), tissue sections were boiled in 10 mM citrate buffer pH 6.0 (Diapath, Martinengo, Italy) for 40 min at 96 °C. They were cooled for 20 min at RT and rinsed in distilled water for 5 min. A permeabilization step with 0.1% (v/v) Triton X-100 (Sigma-Aldrich) was performed at RT for 15 min for Ki67/3β-HSD immunostaining. Non-specific sites were blocked with 5% (w/v) bovine serum albumin (BSA, Sigma-Aldrich) and 5% (v/v) horse serum (Sigma-Aldrich). Slides were then incubated in a humidified environment with primary antibodies (*Supplementary file 1a*), rinsed three times in PBST (PBS with 0.05% Tween-20), and incubated with appropriate secondary antibodies (*Supplementary file 1a*). For Ki67/3β-HSD, a sequential protocol was performed as follows: incubation with anti-Ki67 antibodies overnight at 4 °C, incubation with secondary antibodies coupled to Alexa 594 for 60 min at RT, fixation with 4% PFA for 15 min at RT, and incubation with anti-3β-HSD antibodies directly coupled to Alexa 488 for 90 min at RT. Negative controls were carried out by omitting the primary antibodies. Sections were washed, dehydrated with ethanol, and mounted in Vectashield with Hoechst. Images were acquired on a THUNDER Imager 3D Tissue microscope (Leica) at a 400x magnification. Leydig cell number (3β-HSD$^+$) was normalized to tissue area (cm²).

## Immunohistochemical staining

After deparaffinization and rehydration, endogenous peroxidases were blocked with HP Block (Dako, Les Ulis, France) for 30 min, and non-specific binding sites were blocked with Ultra-V Block solution (Thermo Fisher Scientific) for 10 min at RT. Tissue sections were then incubated overnight at 4 °C with anti-CYP19A1 antibodies (*Supplementary file 1a*). After three 5 min washes in PBS, they were incubated for 10 min at RT with biotinylated polyvalent secondary antibodies (UltraVision Detection System HRP kit, Thermo Fisher Scientific). After three 5 min washes in PBS, a 10 min incubation at RT with streptavidin associated with peroxidase (UltraVision Detection System HRP kit, Thermo Fisher Scientific) was performed. The labeling was revealed after the application of a chromogenic substrate (EnVision FLEX HRP Magenta Chromogen, Dako) for 10 min at RT. A negative control was carried out by omitting the primary antibody.

## RNA extraction and RT-qPCR

### RNA extraction

Total RNA was extracted from testicular samples using the RNeasy Micro kit (Qiagen, Courtabœuf, France) according to the manufacturer's instructions. For *in vitro* cultured tissues, the central necrotic area (16–27% of the explants, *Figure 8—figure supplement 1B–F*) was carefully removed before RNA extraction, so that transcript levels in the healthy part of the samples (i.e. where *in vitro* spermatogenesis occurs) could be measured and compared with *in vivo* controls. To avoid contamination with genomic DNA, extracted RNA was incubated with two units of TURBO DNase (Life Technologies) for 45 min at 37 °C. The amount of the RNA samples was measured with a NanoDrop Spectrophotometer (NanoDrop Technologies, Wilmington, DE, USA) and purity was determined by calculating the ratio of optical densities at 260 nm and 280 nm.

### Reverse transcription

The reverse transcription reaction was performed in a 96-well plate from 1 μg total RNA with 4 μL of qScript cDNA SuperMix 5 X (QuantaBio, Gaithersburg, MD, USA) and ribonuclease-free water to adjust the volume to 20 μL. Reverse transcription was performed according to the following program: 5 min at 25 °C, 30 min at 42 °C, and 5 min at 85 °C. The complementary DNA (cDNA) obtained was diluted 1:10.

### Polymerase chain reaction

cDNA amplifications were carried out in a total volume of 13 μL containing 6 ng of cDNA templates, 6.5 μL of SYBR Green (Thermo Fisher Scientific), and 300 nM of each primer. Specific primers are

listed in **Supplementary file 1b**. Samples were dispensed using the Bravo pipetting robot (Agilent Technologies, Santa Clara, CA, USA). Reactions were performed in 384-well plates (Life Technologies) in a Quant Studio 12 K Flex system. The amplification condition was 20 s at 95 °C followed by 40 cycles (1 s at 95 °C, 20 s at 60 °C) and a final step of denaturation of 15 s at 95 °C, 1 min at 60 °C and 15 s at 95 °C. Melting curves were obtained to ensure the specificity of PCR amplifications. The size of the amplicons was verified by agarose gel electrophoresis (E-gel 4%, Life Technologies). The relative expression level of each gene was normalized to two housekeeping genes as recommended by the MIQE guidelines (**Bustin et al., 2009**): *Gapdh* and *Actb*, which were identified and validated as the most stable and suitable genes for RT-qPCR analysis in mouse testis development (**Gong et al., 2014**). 3β-hydroxysteroid dehydrogenase (*Hsd3b1*), a selective Leydig cell marker, has been used as a normalization factor for the analysis of genes expressed in Leydig cells (**Cacciola et al., 2013**). Data were analyzed using the $2^{-\Delta\Delta Ct}$ method (**Livak and Schmittgen, 2001**).

## Western blot
### Protein extraction
Testes were homogenized in ice-cold RIPA buffer (50 mM Tris HCl, 0.5% cholic acid, 0.1% SDS, 150 mM NaCl) containing a protease inhibitor cocktail (Sigma-Aldrich). For *in vitro* cultured tissues, the central necrotic area was carefully removed before protein extraction.

### Determination of protein concentration by the Bradford method
Total protein concentration was measured in the homogenates. The assay was performed in a 96-well plate in a final volume of 200 µL containing the sample and Bradford's solution (BioRad, Marnes-la-Coquette, France). After a 5 min incubation at RT, the optical density at 595 nm was measured using a Spark spectrophotometer (Tecan, Lyon, France).

### Western blot
Protein extracts (20 µg) were separated on 12% resolving polyacrylamide gels (BioRad) and transferred onto nitrocellulose membranes (BioRad). The membranes were blocked with 5% milk or 5% BSA in TBST (0.2% Tween-20 in Tris Buffered Saline) for 30 min at 37 °C before incubation overnight at 4 °C with the appropriate primary antibody (**Supplementary file 1a**). After washes, membranes were incubated with the appropriate HRP-conjugated secondary antibodies (**Supplementary file 1a**). ECL reagents (BioRad) were used to detect the immunoreactivity. Images were captured with the ChemiDoc XRS + Imaging System (BioRad). Densitometric analyses were performed using Image Lab v5.0 software (BioRad) and protein levels were normalized to β-actin or to the Leydig cell-specific marker 3β-HSD.

### Intratesticular cholesterol levels
Total testicular lipids were extracted from ~10 mg tissue with a Lipid Extraction kit (ab211044; Abcam, Paris, France) according to the manufacturer's instructions. Dried lipid extracts were reconstituted in 200 µL of assay buffer. Intratesticular levels of total, free, and esterified cholesterol were measured with a colorimetric Cholesterol/Cholesteryl Ester assay kit (ab65359; Abcam) according to the manufacturer's instructions.

## Hormonal assays
### Sample preparation for liquid chromatography coupled to mass spectrometry (LC-MS/MS)
Testicular fragments were homogenized in 200 µL (6 d*pp* tissues and *in vitro* matured tissues) or 400 µL (22 d*pp* and 36 d*pp* tissues) of 0.1 M phosphate buffer pH 7.4 with a cocktail of protease inhibitors (Sigma-Aldrich). Total protein concentration was measured as described above.

## LC-MS/MS
The standards for androstenedione, testosterone, dehydroepiandrosterone (DHEA), and their stable labeled isotopes were obtained from Merck. Working solutions were prepared in methanol. Serial dilutions from working solutions were used to prepare seven-point calibration curves and three quality

control levels for all analytes. For the final calibration and quality control solutions, PBS was used for testicular fragments and α-MEM for the medium. The linearity ranges were from 0.05 to 10 ng/mL for androstenedione and testosterone, and from 1 to 200 ng/mL for DHEA. A simple deproteinization was carried out by an automated sample preparation system, the CLAM-2030 (Shimadzu Corporation, Marne-la-Vallée, France) coupled with a 2D-UHPLC-MS/MS system. Once the sample was on board, 30 μL was automatically pipetted in a pre-conditioned tube containing a filter, in which reagents were added, then mixed and filtered. Briefly, the polytetrafluoroethylene (PTFE) filter vial (0.45 μm pore size) was previously conditioned with 20 μL of methanol (Carlo Erba, Val-de-Reuil, France). Successively, 30 μL of sample and 60 μL of a mixture of isotopically labeled internal standards in acetonitrile were added. The mixture was agitated for 120 s (1900 rpm), then filtered by application of vacuum pressure (–60 to –65 kPa) for 120 s into a collection vial. Finally, 30 μL of the extract was injected into the 2D-UHPLC-MS/MS system.

Analysis was performed on a two dimensions ultra-performance liquid chromatograph-tandem mass spectrometer (2D-UHPLC-MS/MS) consisting of the following Shimadzu modules (Shimadzu Corporation): an isocratic pump LC20AD SP, for pre-treatment mode, a binary pump consisting of coupling two isocratic pumps Nexera LC30AD for the analytical mode, an automated sampler SIL-30AC, a column oven CTO-20AC and a triple-quadrupole mass spectrometer LCMS-8060. The assay was broken down into two stages. The first was the pre-treatment where the sample was loaded on the perfusion column. The second step was the elution of the compounds of interest to the analytical column. The deproteinized extract performed by the CLAM-2030 was automatically transferred to the automated sampler, where 30 μL was directly analyzed into the chromatographic system. The LC-integrated online sample clean-up was performed using a perfusion column Shimadzu MAYI-ODS (5 mm L × 2 mm I.D.). The first step consisted of loading the extract on the perfusion column with a mobile phase composed of 10 mM ammonium formate in water (Carlo Erba) at a flow rate of 0.5 mL/min for 2 min. Then the system switched to the analytical step to elute the analytes from the perfusion column to the analytical column to achieve chromatographic separation. During this step, the loading line was washed with propan-2-ol (Carlo Erba) for 3 min. Chromatographic separation was achieved on a Restek Raptor Biphenyl (50 mm L × 3 mm I.D., 2.7 μm) maintained at 40 °C and a gradient of (A) 1 mM ammonium fluoride buffer in water (Carlo Erba) and (B) methanol (Carlo Erba) at a flow rate of 0.675 mL/min as follows: 0.0–2.10 min, 5% (B); 2.1–3.0 min, 5–65% (B); 3.0–4.75 min, 65% (B); 4.75–5.0 min, 65–70% (B); 5.0–6.6 min, 70% (B); 6.6–8.0, 70–75% (B); 8.0–8.5 min, 75–100% (B); 8.50–9.5 min, 100% (B); 9.5–9.6 min, 100–5% (B); 9.6–12.0 min, 5% (B). Detection and quantification were performed by scheduled- Multiple Reaction Monitoring (MRM) using 1ms pause time and 50ms dwell times to achieve sufficient points per peak. The interface parameters and common settings were as follows: interface voltage: 1 kV; nebulizing gas flow: 3 L/min; heating gas flow: 10 L/min; drying gas flow: 10 L/min; interface temperature: 400 °C; desolvation line (DL) temperature: 150 °C; heat block temperature: 500 °C; collision gas pressure 300 kPa. Compound-specific MRM parameters are shown in **Supplementary file 1c**.

## Enzyme-linked immunosorbent assay (ELISA)

Steroids were extracted from testicular homogenates and from organotypic culture media with five volumes of diethyl ether (Carlo Erba). The organic phase was then recovered and evaporated in a water bath at 37 °C. The tubes were stored at –20 °C until use.

Progesterone and estradiol levels were measured in 50 μL of testicular homogenates and in 50 μL of culture media using Cayman ELISA kit (582601 for progesterone and 501890 for estradiol, Cayman Chemical Company, Ann Arbor, MI, USA), according to the supplier's recommendations. The sensitivity limit for the progesterone assay was 10 pg/mL with average intra- and inter-trial variations of 13.9% and 9.6%, respectively. The sensitivity limit for the estradiol assay was 20 pg/mL and the lower limit of detection was 6 pg/mL with average intra- and inter-trial variations of 20.25% and 14.975%, respectively.

## Aromatase activity

After homogenization of ~10–50 mg of tissues in 500 μL of Aromatase Assay Buffer, the aromatase activity was measured with the fluorometric Aromatase (CYP19A) Activity assay kit (ab273306; Abcam) according to the manufacturer's instructions.

## Radioimmunoassay (RIA)

INSL3 was assayed by using RIA kit with $^{125}$I as a tracer (RK-035–27, Phoenix Pharmaceuticals, Strasbourg, France). According to the manufacturer's instructions, analyses were performed on 100 µL of culture media or tissue homogenates. Assay validation was assessed by determining the recovery of expected amounts of INSL3 in samples to which exogenous INSL3 was added. The sensitivity of the INSL3 RIA kit was 60.4 pg/mL and the range of the assay was 10–1280 pg/mL.

## Statistical analyses

Statistical analyses were carried out with GraphPad Prism 8 software (GraphPad Software Inc, La Jolla, CA, USA). Data are presented as means ± SEM. The non-parametric Mann-Whitney test was used to compare *in vitro* cultures and *in vivo* controls (D16 FT *vs* 22 d*pp*, D16 CSF *vs* 22 d*pp*, D30 FT *vs* 36 d*pp*, D30 CSF *vs* 36 d*pp*, D30 FT + hCG *vs* 36 d*pp*, D30 CSF + hCG *vs* 36 d*pp*); cultures of fresh and CSF tissues (6 d*pp vs* 6 d*pp* CSF, D16 FT *vs* D16 CSF, D30 FT *vs* D30 CSF, D30 FT + hCG *vs* D30 CSF + hCG); cultures with or without hCG (D30 FT + hCG *vs* D30 FT, D30 CSF + hCG *vs* D30 CSF). A value of p<0.05 was considered statistically significant.

## Acknowledgements

Data from RT-qPCR and images were obtained on PRIMACEN (http://primacen.crihan.fr), the Cell Imaging Platform of Normandy, IRIB, Faculty of Sciences, University of Rouen, 76821 Mont-Saint-Aignan. The authors also thank the U1096 team for the access to their western blot devices. This work was supported by a PhD grant from Région Normandie (awarded to LM) and by fundings from Ligue Contre le Cancer Comité de Seine-Maritime (awarded to CR), France Lymphome Espoir (Bourse Sacha awarded to CR), Région Normandie and Europe (RIN Steroids awarded to HL) and ANR (ANR-21-CE14-0068, sc-SpermInVitro awarded to NR). The funders had no role in study design, data collection and analysis, decision to publish, or preparation of the manuscript.

## Additional information

### Funding

| Funder | Grant reference number | Author |
| --- | --- | --- |
| Région Normandie | PhD Grant | Laura Moutard |
| Ligue Contre le Cancer | | Christine Rondanino |
| France Lymphome Espoir | Bourse Sacha | Christine Rondanino |
| Région Normandie | RIN Steroids | Hervé Lefebvre |
| Fonds Européen de Développement Régional | RIN Steroids | Hervé Lefebvre |
| Agence Nationale de la Recherche | ANR-21-CE14-0068 | Nathalie Rives |

The funders had no role in study design, data collection and interpretation, or the decision to submit the work for publication.

### Author contributions

Laura Moutard, Conceptualization, Formal analysis, Investigation, Visualization, Methodology, Writing – original draft; Caroline Goudin, Catherine Jaeger, Estelle Louiset, Christelle Delalande, Investigation; Céline Duparc, Tony Pereira, François Fraissinet, Investigation, Methodology; Marion Delessard, Justine Saulnier, Aurélie Rives-Feraille, Validation; Hervé Lefebvre, Funding acquisition; Nathalie Rives, Conceptualization, Supervision, Funding acquisition, Project administration; Ludovic Dumont, Supervision, Project administration, Writing – review and editing; Christine Rondanino, Conceptualization, Supervision, Funding acquisition, Validation, Project administration, Writing – review and editing

**Author ORCIDs**
Laura Moutard https://orcid.org/0000-0003-2412-4377
Catherine Jaeger http://orcid.org/0000-0001-8519-8893
François Fraissinet http://orcid.org/0000-0002-0120-5142
Christine Rondanino https://orcid.org/0000-0002-1100-9401

**Ethics**
All the experimental procedures were approved by the Institutional Animal Care and Use Committee of Rouen Normandy University under the licence/protocol number APAFiS #38239.

Reviewer #1 (Public Review): https://doi.org/10.7554/eLife.85562.4.sa1
Reviewer #2 (Public Review): https://doi.org/10.7554/eLife.85562.4.sa2
Author Response https://doi.org/10.7554/eLife.85562.4.sa3

---

# Additional files

**Supplementary files**
• Supplementary file 1. Detailed tables of antibodies, primers and LC-MS/MS's parameters used. (a) Detailed list of antibodies used in this study HRP: horseradish peroxidase; IF: immunofluorescence; IHC: immunohistochemistry; O/N: overnight; RT: room temperature; WB: western blot. (b) List of PCR primers used in this study. (c) Compound-specific MRM parameters for LC-MS/MS CE: collision energy; DHEA: Dehydroepiandrosterone; MRM: multiple reaction monitoring

• MDAR checklist

• Source data 1. Western Blot for 3β-HSD, CYP17A1, AR, CYP19A1, and FAAH.

**Data availability**
Figure source data contains the numerical data used to generate all the figures. All blots are contained in *Source data 1*.

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
