## [Editor Report · eLife assessment]

This study reports **useful** information on the limits of the organotypic culture of neonatal mouse testes, which has been regarded as an experimental strategy that can be extended to humans in the clinical setting for the conservation and subsequent re-use of testicular tissue. The evidence that the culture of testicular fragments of 6.5-day-old mouse testes does not allow optimal differentiation of steroidogenic cells is **compelling** and should enable further optimizations in the future.

---

## [Referee Report · Reviewer #1 (Public Review)]

In this manuscript, the authors aimed to compare, from testis tissues at different ages from mice in vivo and after culture, multiple aspects of Leydig cells. These aspects included mRNA levels, proliferation, apoptosis, steroid levels, protein levels, etc. A lot of work was put into this manuscript in terms of experiments, systems, and approaches. The technical aspects of this work may be of interest to labs working on the specific topics of in vitro spermatogenesis for fertility preservation.

---

## [Referee Report · Reviewer #2 (Public Review)]

Moutard, Laura, et al. investigated the gene expression and functional aspects of Leydig cells in a cryopreservation/long-term culture system. The authors found that critical genetic markers for Leydig cells were diminished when compared to the in-vivo testis. The testis also showed less androgen production and androgen responsiveness. Although they did not produce normal testosterone concentrations in basal media conditions, the cultured testis still remained highly responsive to gonadotrophin exposure, exhibiting a large increase in androgen production. Even after the hCG-dependent increase in testosterone, genetic markers of Leydig cells remained low, which means there is still a missing factor in the culture media that facilitates proper Leydig cell differentiation. Optimizing this testis culture protocol to help maintain proper Leydig cell differentiation could be useful for future human testis biopsy cultures, which will help preserve fertility and child cancer patients.

Overall, the authors addressed most comments and questions from the previous review. The additional data regarding the necrotic area is helpful for interpreting the quality of the cultures.

The authors did not conduct multiple comparison tests although there are multiple comparisons conducted for a single dependent variable (Fig 2J, Fig 3F, among many others), however, the addition of this multiple comparison is unlikely to change the conclusions of the paper or the figure and, thus is a minor technical detail in this case.

---

## [Author Response]

The following is the authors’ response to the previous reviews

We would like to thank you again for your thorough review of the manuscript. We have taken all comments into account in the revised version of the manuscript. Please find below our detailed responses to your comments.

**Reviewing Editor**
The manuscript has been improved, but there are some remaining issues that need to be addressed, as outlined in the reviewers' comments. In particular, please pay attention to Figures 1A and 2A as they appear to be the same. Moreover, the original gel images for Western blots should be made available given the concerns raised by Reviewer #1.

Thank you for your recommendations. We have carefully considered all comments and made the requested revisions to improve the manuscript.

**Reviewer #1 (Public Review):**
In this manuscript, the authors aimed to compare, from testis tissues at different ages from mice in vivo and after culture, multiple aspects of Leydig cells. These aspects included mRNA levels, proliferation, apoptosis, steroid levels, protein levels, etc. A lot of work was put into this manuscript in terms of experiments, systems, and approaches. The technical aspects of this work may be of interest to labs working on the specific topics of in vitro spermatogenesis for fertility preservation.Second review:The authors should be commended for substantial improvement in their manuscript for resubmission.

Thank you very much for this second review and your help to improve this manuscript.

**Recommendations For The Authors:**
Going forward, the authors would be well-served to put a similar amount of effort on first drafts as well, which would both increase reviewer enthusiasm and reduce reviewer workload to document all the deficiencies! Abstract is much improved, and clearly articulates the point of the study.

We are very grateful for all your constructive comments, which have greatly contributed to the improvement of our manuscript.

1. 54 - replace "could be" with was

“could be” was replaced by “was”

1. 75 - delete "being"

“being” was deleted.

1. 103 - would say "indirectly promotes" since Rhox5 is a transcription factor that presumably activates genes in Sertoli cells whose products then affect neighboring germ cells, either by direct action or by influencing Sertoli cell behavior changes

“indirectly” was added in the sentence.

1. 139, 155, elsewhere - haven't seen dpp italicized before, certainly not the norm

In dpp (days post-partum), “pp” is italicized as it is a Latin word.

1. 265 - delete "found"

“found” was deleted.

1. 263-273 - Is the CYP19 protein referred to encoded by the Cyp19a1 gene (line 263)? Should standardize nomenclature...

The CYP19 protein (aromatase) is indeed encoded by the Cyp19a1 gene. The nomenclature was standardized: “CYP19” was replaced by “CYP19A1” in the entire manuscript.

1. 280 - "homolog" doesn't seem like the right word, as it has a very specific meaning with regards to the evolutionary genetic relatedness of genes. Maybe analog?

“homolog” was replaced by “analog”.

1. 306 - would reword to something like "proportions of seminiferous tubules containing round and elongating spermatids" - the because the tubules don't reach spermatid stages

This sentence was reworded as suggested.

1. 310 - delete "resulted in", unnecessary

“resulted in” was deleted.

1. Why are the images shown in Figures 1A and 2A the same? That seems odd - was that intentional? Curious overall why the data is presented in such a way that it's done twice...

We mistakenly presented immunofluorescence images twice. Duplicate images have been removed. In the modified version of this manuscript, Figure 1A shows 3β-HSD immunofluorescence staining in cultures of fresh testicular tissues and in their in vivo counterparts while Figure 1 – figure supplement 1A (not Figure 2A) shows 3β-HSD immunofluorescence staining in cultures of frozen/thawed testicular tissues.

1. In all the western blots, the cropping is done awfully close to the bands - why is this? Can full gels be shown in a Supplement? And especially in the westerns in Fig. 5C, esp for CYP17A1, the cropping is unacceptable. This reviewer is wondering whether this is an oversight, or whether there is another band below that one that is being masked? Again, should show whole blot for transparency and to ensure Rigor and Reproducibility.

Full gels are shown in the Supplementary File 2. For CYP17A1, we have shown that only one band of the expected molecular weight is obtained with the antibody (Please see photo below). After this verification, the nitrocellulose membranes were cut at the 55 kDa molecular weight band in order to reveal CYP17A1 expression in the upper part of the membranes and the protein used for normalization in the lower part of the membranes.

1. For all figures, wondering why the font sizes are so disparate? This will need to be addressed before publication so it looks more professional.

All figures have been reworked as requested.

**Reviewer #3 (Public Review):**
Moutard, Laura, et al. investigated the gene expression and functional aspects of Leydig cells in a cryopreservation/long-term culture system. The authors found that critical genetic markers for Leydig cells were diminished when compared to the in-vivo testis. The testis also showed less androgen production and androgen responsiveness. Although they did not produce normal testosterone concentrations in basal media conditions, the cultured testis still remained highly responsive to gonadotrophin exposure, exhibiting a large increase in androgen production. Even after the hCG-dependent increase in testosterone, genetic markers of Leydig cells remained low, which means there is still a missing factor in the culture media that facilitates proper Leydig cell differentiation. Optimizing this testis culture protocol to help maintain proper Leydig cell differentiation could be useful for future human testis biopsy cultures, which will help preserve fertility and child cancer patients.Overall, the authors addressed most comments and questions from the previous review. The additional data regarding the necrotic area is helpful for interpreting the quality of the cultures. The authors did not conduct a multiple comparison tests although there are multiple comparisons conducted on for a single dependent variable (Fig 2J, Fig 3F, among many others), however, the addition of this multiple comparison is unlikely to change the conclusions of the paper or the figure and, thus is a minor technical detail in this case.

Thank you very much for this second review and your help to improve this manuscript.